# Data Augmentation for Medical Imaging: Counterfactual Simulation of Acquisition Parameters via Conditional Diffusion Model

**Pedro Morão**[1]             PEDRO.MORAO@TECNICO.ULISBOA.PT
[1] *Instituto Superior Técnico - Universidade de Lisboa, Lisboa, Portugal*

**Yasna Forghani**[2] (iD)         YASNA.FORGHANI@RESEARCH.FCHAMPALIMAUD.ORG
[2] *Digital Surgery LAB, Breast Unit, Champalimaud Foundation, Lisboa, Portugal*

**Nuno Loução**[2] (iD)          NUNO.LOUCAO@RESEARCH.FCHAMPALIMAUD.ORG
**Pedro Gouveia**[2,3] (iD)        PEDRO.GOUVEIA@FUNDACAOCHAMPALIMAUD.PT
[3] *Faculdade de Medicina, Universidade de Lisboa, Lisboa, Portugal*

**Mário A. T. Figueiredo**[1,4] (iD)   MARIO.FIGUEIREDO@TECNICO.ULISBOA.PT
[4] *Instituto de Telecomunicações, Lisboa, Portugal*

**João Santinha**[2,3] (iD)        JOAO.SANTINHA@RESEARCH.FCHAMPALIMAUD.ORG

**Editors:** Accepted for publication at MIDL 2025

## Abstract

Deep learning (DL) models in medical imaging face challenges in generalizability and robustness due to variations in image acquisition parameters (IAP). In this work, we introduce a novel method using conditional denoising diffusion generative models (cD-DGMs) to generate counterfactual medical images that simulate different IAP without altering patient anatomy. We demonstrate that using these counterfactual images for magnetic resonance (MR) data augmentation can improve segmentation accuracy in out-of-distribution settings, enhancing the overall generalizability and robustness of DL models across diverse imaging conditions. Our approach shows promise in addressing domain and covariate shifts in medical imaging. The code is publicly available at https://github.com/pedromorao/Counterfactual-MRI-Data-Augmentation

**Keywords:** Denoising Diffusion Generative Models, Data Augmentation, MRI, Medical Imaging, Generalizability

## 1. Introduction

Deep learning (DL) models in medical imaging continue to face generalizability and robustness challenges. This is specially relevant given the variability of imaging devices and their ability to change image acquisition parameters. While data augmentation has been widely used to improve the performance of DL models in various fields, current augmentation techniques do not easily replicate domain, population, and covariate shifts that arise from variations in medical image scanners, acquisition settings, and patient populations. As variations in scanners and acquisition settings should only produce changes in the image *style*, style transfer has been proposed as a possible solution to harmonize images across different acquisition settings and scanners (Karras, 2019; Zhu et al., 2017). However, those methods usually work by mapping a source to a target domain on a pairwise basis. That approach

thus leads to combinatorially growing numbers of possible combinations that exponentially increase as new scanners and acquisition protocols emerge.

Invariant-based methods, like the one proposed by Arjovsky et al. (2019), offer a promising solution to mitigate performance drops under domain and covariate shifts. However, those methods often require detailed information about the environments in which the data were acquired, as well as known clinical outcomes. Advances in image generation and modification techniques could be leveraged to synthesize new images (Fernandez et al., 2022; Usman Akbar et al., 2024), further enforcing invariance during training. Several studies have investigated generative methods for counterfactual image generation (Ribeiro et al., 2023; Sanchez and Tsaftaris, 2022; Mertes et al., 2022; Konz et al., 2024). In particular, Ribeiro et al. (2023), Mertes et al. (2022), and Konz et al. (2024) explored these methods in medical imaging, evaluating their ability to simulate variations in patient demographics, MRI sequences, and anatomical structures. However, no prior work has investigated the use of generative models to produce counterfactual images by simulating different acquisition settings while preserving anatomical structures. Thus, exploring how such techniques may improve DL models' robustness and generealizabily is a promising avenue. Towards this goal, this study investigates the following open questions:

**Q1:** Can we develop a generative model capable of counterfactually modifying medical images, in particular MRI, simulating different image acquisition settings?

**Q2:** Can the counterfactual images *fool* a classifier trained to accurately predict the image acquisition parameters from the pixel data?

**Q3:** Does training a segmentation model with counterfactually modified images increase performance on out-of-distribution samples?

Our work introduces a novel method for creating counterfactuals from existing data using conditional denoising diffusion generative models (cDDGMs). Our approach simulates the acquisition of magnetic resonance (MR) images across different scanners and image acquisition parameters (IAP). By incorporating IAP as conditioning context for the denoising diffusion generative model (DDGM), we are able to alter images without affecting the underlying patient anatomy.

We evaluate the effectiveness of the generated counterfactual IAP images using metrics such as the Fréchet inception distance (FID), structural similarity index metric (SSIM), and maximum mean discrepancy (MMD). Additionally, we assess the ability of these images to mislead a multi-task model trained to predict the IAP from MR images. Finally, we examine the impact of using these counterfactual images for data augmentation on the generalizability of DL segmentation models, focusing on both in-distribution (ID) and out-of-distribution (OOD) scenarios.

The main contributions of this study can be summarized as follows:

- We explore and demonstrates the feasibility of using cDDGM to generate counterfactually IAP modified medical images.

- We assess the impact on generalizability when using the proposed cDDGM as an IAP data augmentation method for training segmentation models.

- We approach the stated open questions (Q1-3) by conducting experiments in a public dataset and we open-source the code used in our experiments, allowing further testing and extension of the proposed method by the research community.

## 2. Materials and Methods

### 2.1. Dataset

We used the Duke-Breast-Cancer-MRI dataset (Saha et al., 2018) to train and evaluate our deep generative model and to perform the different experiments. The dataset comprised pre-contrast dynamic contrast-enhanced breast MRIs from 922 patients, with 100 patients (29 acquired with Siemens scanners and 71 acquired with GE scanners) also containing breast tissue segmentation masks. Details pertaining to data normalization and pre-processing can be found in Section A.1. More information regarding data partitioning for the different experiments are provided in Section 3.1.3.

### 2.2. Conditional Denoising Diffusion Generative Model

Our proposed cDDGM is developed to modify MR images through the simulation of their acquisition with counterfactual IAP. The proposed cDDGM architecture is based on the DDPM architecture (Ho et al., 2020), using a conditional U-Net as the noise estimation model which learns to reverse a Markovian diffusion process by gradually denoising an image, starting from pure-noise. Additionally, inspired by the U-Net design from latent diffusion models (Rombach et al., 2022), the proposed conditional U-Net architecture also incorporates cross-attention mechanisms, which enhance the model's ability to effectively handle conditioning contexts that are more complex than simple image classes.

Our U-Net architecture consists of six downsampling layers (number of channels per layer: 64, 64, 128, 128, 256, 256), one middle layer, and six upsampling layers, with each layer containing two residual convolutional blocks. Cross-attention blocks are included on the third and fifth of the downsampling layers, on the middle layer, and the corresponding positions in the upsampling layers. While adding more cross-attention blocks could improve the model's performance, it also significantly increases the computational cost, particularly if added to earlier layers of the U-Net. The conditioning is performed by adding the IAP embedding to the time embeddings and incorporating it through the cross-attention blocks. This "hybrid" conditioning approach, which combines adding the condition embedding to the time embeddings and cross-attention blocks, is similar to what is used by Pinaya et al. (2022).

The model was trained using the simplified loss

$$\mathcal{L}(\theta) := \mathbb{E}_{t,\mathbf{x}_0,\epsilon\sim N(0,\mathbf{I})} \left[ \left\| \epsilon - \epsilon_\theta(\sqrt{\bar{\alpha}_t}\mathbf{x}_0 + \sqrt{1-\bar{\alpha}_t}\epsilon, c, t) \right\|^2 \right], \tag{1}$$

where $t$ is a timestep, $\mathbf{x}_0$ represent the original image; $c$ are a set of IAP used for conditioning, $\bar{\alpha}_t = \prod_{s=1}^{t} \alpha_s$, with $\alpha_t = 1 - \beta_t$, and $\beta_t$ corresponding to the forward process variances at step $t$. This loss function was adapted for the conditional training scenario, allowing the model $\epsilon_\theta$ to receive the IAP conditioning as input but still work in an unconditional setting without $c$.

To condition across the multiple classes, corresponding to the different IAP, we selected the classifier free-guidance (CFG) method (Ho and Salimans, 2022), as it enables controlling the strength of the alignment with the conditional context through a guidance scale parameter, eliminating the need for an additional classifier, as opposed to classifier guidance.

The training algorithm for the cDDGM is equal to the original DDPM training algorithm (Ho et al., 2020), except the model is conditioned on the IAP with a conditional dropout of 15%. The algorithm to counterfactually modify images and simulate their acquisition with other IAP is shown in Algorithm 1. For the diffusion process, we use 1000 steps of the original DDPM sampler with a cosine noise scheduler. Initially, noise is added to the original image $\mathbf{x}_0$ until we reach $t = steps$, then we use the CFG method (Ho and Salimans, 2022) to denoise the image from $t = steps$ back to $t = 0$, now conditioning the image on a new set of IAP, $c_{new}$, and controlling the guidance scale with a parameter $w$. After denoising $\mathbf{x}_{steps}$, we return the modified $\mathbf{x}_0$ with its IAP changed.

---

**Algorithm 1:** IAP modification algorithm using CFG. $t$: timestep; $\mathbf{x}_0$: original image; $c_{new}$: new set of IAP; $\mathbf{x}_t$: resulting images after step $t$; $\alpha_t = 1 - \beta_t$; $\beta_t$: forward process variances at step $t$; $\bar{\alpha}_t = \prod_{s=1}^{t} \alpha_s$.

---

$\mathbf{z} \sim \mathcal{N}(0, \mathbf{I})$
$\mathbf{x}_{steps} = \sqrt{\bar{\alpha}_{steps}}\mathbf{x}_0 + \sqrt{1 - \bar{\alpha}_{steps}}\mathbf{z}$
**for** $t = steps, \cdots, 0$ **do**
    $\mathbf{z} \sim \mathcal{N}(0, \mathbf{I})$ if $t > 0$, else $\mathbf{z} = 0$
    $\tilde{\epsilon}_t = (1 - w)\epsilon_\theta(\mathbf{x}_t, t) + w\epsilon_\theta(\mathbf{x}_t, c_{new}, t)$
    $\mathbf{x}_{t-1} = \frac{1}{\sqrt{\alpha_t}}(\mathbf{x}_t - \frac{1-\alpha_t}{\sqrt{1-\bar{\alpha}_t}}\tilde{\epsilon}_t) + \sigma_t\mathbf{z}$
**end**
**return** $\mathbf{x}_0$

---

This model was trained with a batch size of 32 over 15 epochs. The Adam optimizer was used with a learning rate of $10^{-4}$ and weight decay of $10^{-3}$. Additional training details are provided in Section 3.1.3

## 3. Experiments

### 3.1. Experiments and Metrics to Evaluate Counterfactual Data Augmentation

After training the cDDGM, we counterfactually modified the original images by altering their IAP. To achieve this, we stopped the forward diffusion process at an early stage, when the perturbed image's IAP distributions would overlap, and then reversed the diffusion process while conditioning the image on a different set of IAP. This approach is similar to that of Meng et al. (2022), but we employ a conditional model. We explore the impact of varying the number of reverse diffusion steps and adjusting the CFG's scale parameter on the resulting counterfactual images. To generate the counterfactual version of the input image, a set of IAP from a different manufacturer was randomly selected and used to modify it.

To evaluate the proposed method, we used several image quality and generative metrics, including the structural similarity metric (SSIM), Fréchet inception distance (FID),

and maximum mean discrepancy (MMD). Additionally, inspired by previous work from Konz and Mazurowski (2024), we developed an IAP prediction models to assess whether cDDGM's counterfactual images could "fool" the predictor into classifying them with the counterfactual IAP rather than the original one. We evaluated the IAP predictor model performance using top-1 accuracy for categorical IAP and mean squared error for continuous IAP; further details about this model are provided in Section 3.1.1. Moreover, counterfactual image generation was also assessed through the use of counterfactual prediction gain (Nemirovsky et al., 2020).

Finally, since the developed cDDGM was trained to perform changes in tissue contrast based on the IAP, without changing the anatomy, we then used the IAP counterfactual images as data augmentation samples and assessed the effect on the performance of the segmentation models in the two scenarios presented in Section 3.1.3. The segmentation models are described in 3.1.2. We assessed the impact of the counterfactual data augmentation in the segmentation models' using the Dice-Sørensen coefficient and accuracy for each different breast tissue present in the segmentation masks.

### 3.1.1. IMAGE ACQUISITION PARAMETERS PREDICTION MODEL

Following the model proposed by Konz and Mazurowski (2024), a ResNet-18 (He et al., 2016) was modified to predict 7 image acquisition parameters through the final fully-connected layer. These 7 MRI acquisition parameters change contrast in a non-linear manner, impacting DL segmentation models, as existing harmonization and normalization methods fail to fully compensate for these variations.

The four continuous ($M = 4$) IAP - Flip Angle (FA), Slice Thickness (ST), Echo Time (TE), and Repetition Time (TR) - are predicted directly using a single unit for each of them in our network's output layer. The three categorical ($K = 3$) IAP considered - Scanner Manufacturer (SM), Field Strength (FS), and Scan Options (SO) - are converted into one-hot encoding each with a different number of possible values/categories. For the categorical variables, with $C_k$ ($k = 1, \cdots, K$) denoting the number of categories in each categorical variable, the final layer, has a total width of $\sum_{k=1}^{K} C_k + M$.

The training of the IAP model involved a multi-task learning approach with the combination of loss functions for the categorical (weighted-cross-entropy losses, $\mathcal{L}_{WCE_k}$) and continuous IAP (mean squared error losses, $\mathcal{L}_{MSE}$):

$$\mathcal{L}_{IAP} = \sum_{k=1}^{K} \mathcal{L}_{WCE_k}(\hat{y}, y) + \sum_{m=1}^{M} \mathcal{L}_{MSE}(\hat{y}, y). \tag{2}$$

The IAP prediction model was trained using a batch size of 512 over 200 epochs. The Adam optimizer was used with a learning rate of $10^{-5}$ and a weight decay parameter of $10^{-4}$.

### 3.1.2. BREAST TISSUE SEGMENTATION MODEL

For the breast tissue segmentation, a U-Net (Ronneberger et al., 2015) with residual blocks to enable better gradient back-propagation and facilitate the optimization process, to segment MRI images into 3 different labels fat, fibroglandular tissue (FGT) and background.

The segmentation models were trained using the Adam optimizer with a learning rate of 0.002, a weight decay of 0.001, and a batch size of 256. Early stopping was applied to determine the optimal stopping point during training. The number of channels per layer was 32, 64, 128, 256, 512, and 512.

### 3.1.3. Additional Details on IAP, cDDGM, and Segmentation Model Training

We used images from the 822 patients without breast segmentations to train the cDDGM and IAP models. The training of the segmentation models used the images and corresponding breast tissue segmentation masks of the remaining 100 patients, while considering different scenarios: (1) mix of images from different manufacturers available for training; (2) images from only one manufacturer available for training.

An iterative method was used to split the images into training, validation, and test sets, ensuring that different combinations of IAP were equally represented across all sets. Additionally, the training/validation/test splitting procedure ensured that images from the same patient were not included in different sets.

For the subset without segmentations, the dataset was split into 75% for training, 10% for validation, and 15% for testing. In the subset with segmentations, 75% of the data was used for training and 25% used for validation. Due to the limited number of patients with segmentations, the validation set was also used as the test set to evaluate the segmentation model in an ID setting. All OOD images were used as the test set in the OOD evaluation, as they were not included in the training process.

## 4. Results and Discussion

The guidance scale and number of steps of the proposed cDDGM were optimized for counterfactual IAP modification. As shown in Tables 4 and 5 of Appendix A.3, increasing these hyperparameters leads to deteriorated image quality and generative metrics, such as FID, $\text{SSIM}_{orig. and mod.}$, and MMD, while simultaneously improving predictions of the IAP used to counterfactually modify the images. Consequently, to generate counterfactual IAP samples for data augmentation and to evaluate its impact on segmentation model performance, we selected a guidance scale of 3 and 50 steps. This configuration strikes a balance between preserving image quality and achieving effective IAP prediction.

The performance of the IAP prediction model is summarized in Table 1. We see that the IAP prediction model captures with very good accuracy the IAP of the test dataset. Considering the ranges of each continuous variable (FA: [7°-12°]; ST: [1.1mm-2.5mm]; TE:[1.250ms-2.756ms]; TR: [3.540ms-7.395ms]), the IAP prediction models was able to estimate all variables with low MSE, except ST, for which the MSE was relatively higher ($\sim$ 5-12%).

The proposed method also shows the ability to generate counterfactuals MRIs, which demonstrated the ability to improve counterfactual prediction gains for manufacturer, scanner manufacturer models, and field strength, as show in Table 2.

Table 3 presents the segmentation accuracies for background, fat, and FGT, along with the mean Dice scores for models trained using images from GE and Siemens MRI scanners. This table includes results for in-distribution (ID) settings (e.g., trained on GE, applied to GE; trained on Siemens, applied to Siemens) and out-of-distribution (OOD) settings

Table 1: Model prediction performance for all IAP on the Test Set. An upward arrow indicates that a higher value is better, and vice versa.

| Image acquisition parameter (IAP) | Top-1 pred. acc. (%) ↑ | Pred. MSE ↓ |
|---|---|---|
| Manufacturer Model | 98.9 | NA |
| Field Strength | 99.2 | NA |
| Scan Options | 99.9 | NA |
| Flip Angle ($^o$) | NA | 0.080 |
| Slice Thickness (mm) | NA | 0.133 |
| TE (ms) | NA | 0.005 |
| TR (ms) | NA | 0.046 |

Table 2: Counterfactual prediction gains for the categorical IAPs. The range for counterfactual prediction gain is [0, 1] with an higher prediction gain indicating more improvement.

| | Manufacturer | Scanner | Field Strenght |
|---|---|---|---|
| Counterfactual Prediction Gain | 0.372 | 0.568 | 0.254 |

(e.g., trained on GE, applied to Siemens; trained on Siemens, applied to GE). Additionally, we tested the use of the cDDGM as a data augmentation method for each setting, by generating modified images simulating the acquisition of the training images using set of IAP from images of a different manufacturer.

The results indicate that using IAP counterfactual images yields slight improvements in segmentation accuracy for the background and FGT, as well as an enhanced Dice score for fat in an ID setting with GE scanners. Similar improvements were observed for fat and FGT when using Siemens scanners. In these ID scenarios, we did not expect larger segmentation performance improvements, as the counterfactual data augmentation generated by the proposed cDDGM produces OOD samples.

In OOD settings, when the model was trained with GE images, the inclusion of IAP counterfactual images positively impacted the accuracies for background and fat, as well as the mean and fat Dice scores. Despite statistically significant improvements of mean and fat Dice, the same improvement was not observed for FGT. FGT is characterized by having smaller dimensions, a more variable intensity distribution, patients with very little or no FGT, and in some case less distinguishable boundaries from surrounding structures compared to fat, making segmentation harder and more uncertain. In the OOD setting, when the model was trained on Siemens images and applied to GE images, the IAP counterfactual model provide statistically significant improvements on segmentation accuracy for fat and FGT, along with enhancing the fat, FGT, and mean Dice scores.

Figure 1 showcases several examples of breast MRIs from the test set, along with corresponding ground truth tissue masks and DL segmentation predictions without and with IAP counterfactual images used as data augmentation in the two OOD settings previously

Table 3: Segmentation performance in ID (equal manufacturers in training and testing) and OOD (different manufacturers in training and testing) cases with and without counterfactual IAP data augmentation. CF: counterfactual; Acc.: accuracy; Backgr.: background; FGT: fibrogladular tissue; Manuf.: manufacturer; * - $p < 0.05$.

| Training Manuf. | Testing Manuf. | Acc. Backgr. ↑ | Acc. Fat ↑ | Acc. FGT ↑ | Dice Fat ↑ | Dice FGT ↑ | Mean Dice ↑ |
|---|---|---|---|---|---|---|---|
| Both | Both | 99.4 | 93.6 | 90.3 | 0.929 | 0.716 | 0.840 |
| GE | GE | 99.4 | **96.6** | 88.6 | 0.949 | **0.758** | **0.863** |
| GE + CF Siemens | GE | **99.5** | 96.5 | **89.4*** | **0.950** | 0.757 | **0.863** |
| GE | Siemens | 98.3 | 94.0 | **62.3** | 0.860 | **0.555*** | 0.730 |
| GE + CF Siemens | Siemens | **99.1*** | **94.2** | 61.7 | **0.889*** | 0.536 | **0.739*** |
| Siemens | Siemens | **99.4** | 91.5 | 58.7 | **0.866** | **0.588*** | **0.746*** |
| Siemens + CF GE | Siemens | 99.1 | **92.3*** | **60.2*** | 0.863 | 0.571 | 0.737 |
| Siemens | GE | **98.9** | 89.3 | 59.7 | 0.886 | 0.549 | 0.742 |
| Siemens + CF GE | GE | 98.8 | **91.8*** | **67.7*** | **0.896*** | **0.553*** | **0.750*** |

mentioned. In Figure 1-A, we observe that the DL segmentation model without cDDGM data augmentation has a propensity to incorrectly classify background areas (black) in the chest wall (top image) and liver (third image), where the model with cDDGM data augmentation was able to reduce these errors. As for Figure 1-B, we see several holes in the breast tissue masks of the first and third predictions using the DL segmentation model trained without cDDGM data augmentation that are reduced when the proposed data augmentation method is used. Figure 2 in the Appendix A demonstrates that for both ID scenarios.

Our work is limited by the lack of diversity in MRI scanner manufacturers and the dataset size (e.g., of the 100 patients containing breast tissue segmentations, only 29 patients were acquired in Siemens scanners). Nevertheless the use of the proposed cDDGM for counterfactual MRI data augmentation yielded promising results, demonstrating its potential to improve generalizability and robustness of DL models in medical imaging.

## 5. Conclusions

In this work, we demonstrated that integrating image acquisition parameters counterfactual images using conditional denoising diffusion generative models can enhance the generalizability and robustness of deep learning models in medical imaging. The generated counterfactual images successfully misled the image acquisition parameters prediction model into predicting the intended counterfactual parameters. Moreover, using these images for data augmentation led to slight improvements in segmentation accuracy, particularly in out-of-

Figure 1: Comparison between the ground truth (True) and DL breast segmentation models trained without data augmentation (Pred.) and with data augmentation using cDDGM (Pred.Aug.), in out-of-distribution settings. (A) results of models trained in GE when applied to Siemens MRIs. (B) results of models trained in Siemens when applied to GE MRIs. Blue - Fat mask; Orange - FGT mask.

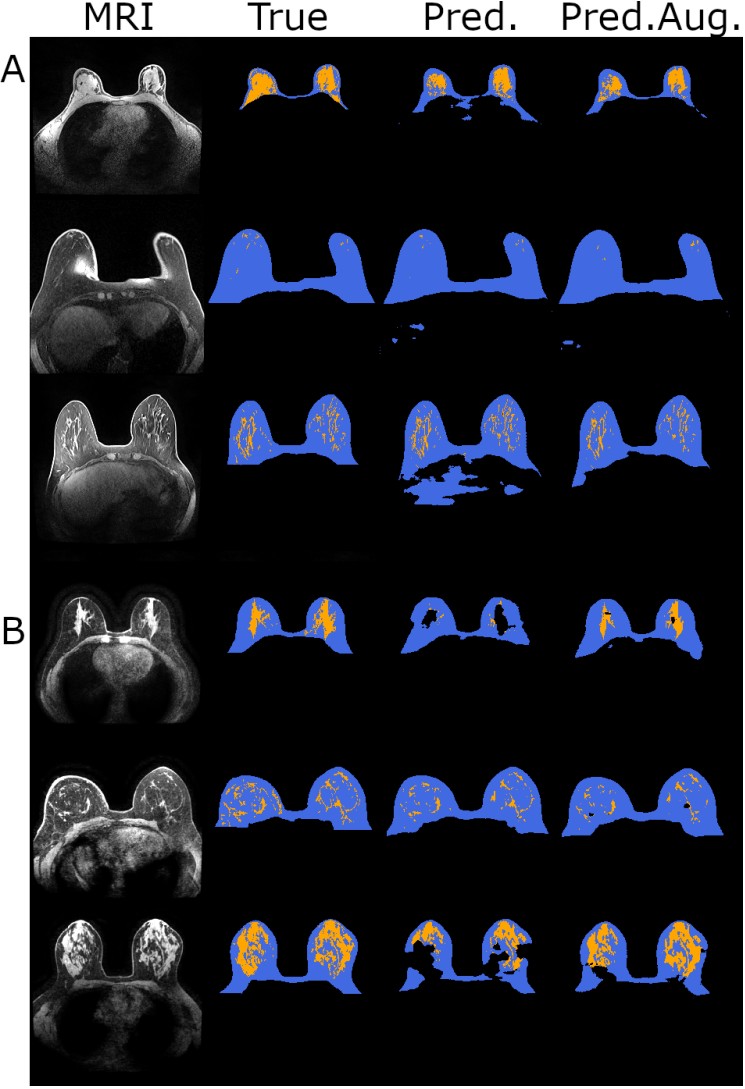

distribution settings, thereby improving the generalizability of deep learning models across diverse medical imaging conditions.

## Acknowledgments

This study received support from the "Health from Portugal - Agenda Mobilizadora para a Inovação Empresarial" project, funded by Plano de Recuperação e Resiliência português under grant agreement No C644937233-00000047.

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

## Appendix A. Appendix

### A.1. Data Normalization and Preprocessing

The Duke-Breast-Cancer-MRI dataset comprises multiple 3D and 4D MRI sequences. Since each sequence is associated with only one set of IAP, pairwise supervised image modification techniques are not applicable. Following the approach of Konz and Mazurowski (2024), we focused on the 3D pre-contrast phase of 4D dynamic contrast-enhanced sequences. In 100 patients, this phase included corresponding 3D fat and fibroglandular tissue segmentations, enabling us to evaluate the impact of the proposed cDDGM model as a data augmentation technique for DL segmentation models.

Although the selected phase represents a 3D volume, due to the size of the cDDGM model and hardware constraints, we developed and evaluated our model using 2D slices extracted from the 3D volumes. The first and last 20 slices were discarded, as they typically contained more noise and lacked relevant information.

We performed image normalization by resizing the images to 224x224 to ensure a fixed size and to accelerate model training and optimization. Although more complex models, such as Latent Diffusion Models, could handle larger images, they would require additional training of encoder and decoder networks to obtain smaller latent space in which the diffusion process would be executed. Moreover, the encoder and decoder would also need to preserve IAP-related information to ensure that the latent representation would still contain such information.

Image intensity values were normalized using percentile normalization, setting the 10th percentile to 0 and the 99th percentile to 1, without clipping values. The lower percentile was adjusted to a higher value due to the large number of low-intensity voxels in the background and thoracic cavity, which are not particularly relevant for the cDDGM or breast tissue segmentation model.

To normalize the IAP, categorical features were one-hot encoded, and numeric features were normalized by dividing by the maximum value in the dataset. This approach was chosen over min-max normalization to create a gap from 0 to the ratio of $value_{min}/value_{max}$, allowing the model to use 0 as the unconditional value. And since $value_{min} \neq 0$, this is always achievable.

### A.2. Computational Resources and Training Setup

The training of all deep learning models was carried out using Pytorch (Paszke et al., 2019), MONAI Core (Cardoso et al., 2022), and MONAI Generative (Pinaya et al., 2023) libraries.

The training processes were conducted on a single NVIDIA A6000 GPU with 48 GB of memory.

The IAP prediction model training and testing phases combined took approximately 5 hours.

For the cDDGM model, the training took 9 hours and the testing the IAP modifications applied to the test set took from 2 to 7 hours for configuration of steps and guidance scales, varying with the number of steps.

Data augmentation was performed, with processing times ranging from 3 to 10 hours, depending on the manufacturer and the number of steps specified.

The segmentation model training, which followed the data augmentation, typically took from 30 minutes up to 1 hour, with the models trained on the larger GE subset requiring more time.

### A.3. cDDGM Optimization

The guidance scale and number of steps of the proposed cDDGM were optimized for counterfactual IAP modification. The performance of the model with the different hyperparameters was assessed using generative and similarity metrics, shown in Table 4, along with the IAP prediction model performance, shown in Table 5. Within Table 4, $\text{SSIM}_{orig. \ and \ mod.}$ was computed between between pairs original images and corresponding modified images, while $\text{SSIM}_{shuff. \ and \ mod.}$ was obtained between images with the set of IAP used to modify the images and the images modified by our cDDGM using those IAP as conditioning. A low $\text{SSIM}_{shuff. \ and \ mod.}$ value is expected as the images being compared were not from the same patients.

Table 4: cDDGM performance metrics on the IAP modification. FID: Fréchet inception distance, SSIM: Structural similarity index metric, MMD: Maximum mean discrepancy. $\text{SSIM}_{orig. \ and \ mod.}$ represents structural similarity index between original images and corresponding modified images. $\text{SSIM}_{shuff. \ and \ mod.}$ represents the structural similarity index between images from which the IAP were originally obtained and the images modified by our cDDGM using those IAP as conditioning - importantly, the images being compared were not from the same patients. An upward arrow indicates that a higher value is better, and vice versa.

| Hyperparameters | FID $\downarrow$ | $\text{SSIM}_{orig. \ and \ mod.}\uparrow$ | $\text{SSIM}_{shuff. \ and \ mod.}\downarrow$ | MMD $\downarrow$ |
|---|---|---|---|---|
| Without cDPPM | 0 | 1.000 | 0.258 | 0 |
| # steps: 25; gs: 3 | 0.416 | 0.742 | 0.284 | $0.010\text{x}10^{-3}$ |
| # steps: 25; gs: 5 | 0.501 | 0.709 | 0.277 | $0.016\text{x}10^{-3}$ |
| # steps: 25; gs: 7 | 0.590 | 0.689 | 0.272 | $0.022\text{x}10^{-3}$ |
| # steps: 50; gs: 3 | 0.513 | 0.657 | 0.287 | $0.016\text{x}10^{-3}$ |
| # steps: 50; gs: 5 | 0.606 | 0.630 | 0.281 | $0.025\text{x}10^{-3}$ |
| # steps: 50; gs: 7 | 0.702 | 0.613 | 0.276 | $0.037\text{x}10^{-3}$ |
| # steps: 75; gs: 3 | 0.573 | 0.606 | 0.288 | $0.029\text{x}10^{-3}$ |
| # steps: 75; gs: 5 | 0.669 | 0.583 | 0.283 | $0.036\text{x}10^{-3}$ |
| # steps: 75; gs: 7 | 0.774 | 0.566 | 0.279 | $0.049\text{x}10^{-3}$ |

In the table 4 and 5, the row 'Without cDDGM' represents the baseline case where the model is not being applied to the images so the IAP are just being shuffled randomly for the computation of $\text{SSIM}_{shuff. \ and \ mod.}$ and the prediction of the IAP.

Table 5 shows that increasing the guidance scale and the amount of noise - through additional forward diffusion steps applied to the original image - enhances the proposed model's ability to predict the IAP used to generate counterfactual images. Regarding the MSE obtained for the continuous variables, we can observe that these results represent small error when compared with the range of value of each continuous variable presented in

Table 5: Model Prediction Performance for all IAP on the Test Set. An upward arrow indicates that a higher value is better, and vice versa.

| Hyperparam. | MM Top-1 pred. acc. (%) ↑ | FS Top-1 pred. acc. (%) ↑ | SO Top-1 pred. acc. (%) ↑ | FA Pred. MSE ↓ | ST Pred. MSE ↓ | TR Pred. MSE ↓ | TE Pred. MSE ↓ |
|---|---|---|---|---|---|---|---|
| Without cDPPM | 22.8 | 49.8 | 28.0 | 0.440 | 0.260 | 1.046 | 0.481 |
| # steps: 25 gs: 3 | 77.7 | 66.3 | 87.8 | 0.289 | 0.198 | 0.293 | 0.070 |
| # steps: 25 gs: 5 | 82.4 | 71.7 | 92.8 | 0.292 | 0.192 | 0.249 | 0.055 |
| # steps: 25 gs: 7 | 83.4 | 76.0 | 93.0 | 0.289 | 0.189 | 0.237 | 0.058 |
| # steps: 50 gs: 3 | 85.8 | 76.9 | 96.0 | 0.282 | 0.190 | 0.207 | 0.038 |
| # steps: 50 gs: 5 | 87.2 | 83.1 | 96.3 | 0.284 | 0.185 | 0.186 | 0.040 |
| # steps: 50 gs: 7 | 88.3 | 87.0 | 96.6 | 0.275 | 0.180 | 0.179 | 0.042 |
| # steps: 75 gs: 3 | 87.1 | 82.8 | 97.0 | 0.285 | 0.189 | 0.183 | 0.034 |
| # steps: 75 gs: 5 | 88.5 | 88.5 | 96.5 | 0.273 | 0.181 | 0.174 | 0.039 |
| # steps: 75 gs: 7 | 89.3 | 90.4 | 96.6 | 0.257 | 0.172 | 0.172 | 0.042 |

Section 4. However, this comes at the cost of reduced image quality, as indicated by higher FID scores, lower SSIM$_{orig.\ and\ mod.}$ values, and increased MMD, as observed in Table 4. Specifically, the decline in SSIM between the original and modified images suggest that higher guidance scales and more diffusion steps lead to greater loss of the original image anatomical structure. Despite these changes, the MMD remains very small, indicating that the modified images stay close to the desired distribution after IAP modification. Additionally, the SSIM between the images of different patients from which the IAP were extrated to condition the image modification ($shuffled$) and the modified images remains low, suggesting that the proposed cDDGM is not altering the structure to resemble that of the $shuffled$ reference image.

Figure 2 shows several examples of breast MRIs from the test sets, along with corresponding ground truth tissue masks and segmentation predictions without and with IAP counterfactual images used as data augmentation in the two ID settings. In both scenarios, A and B, corresponding to training and inference on images from GE and Siemens, respectively, we can observe that the DL segmentation models with and without cDDGM data augmentation perform similarly and are able approximate the ground truth.

Figure 2: Comparison between the ground truth (True) and DL breast segmentation models trained without data augmentation (Pred.) and with data augmentation using cDDGM (Pred.Aug.), in in-distribution settings. (A) results of models trained in GE when applied to GE MRIs. (B) results of models trained in GE when applied to GE MRIs. Blue - Fat mask; Orange - FGT mask.

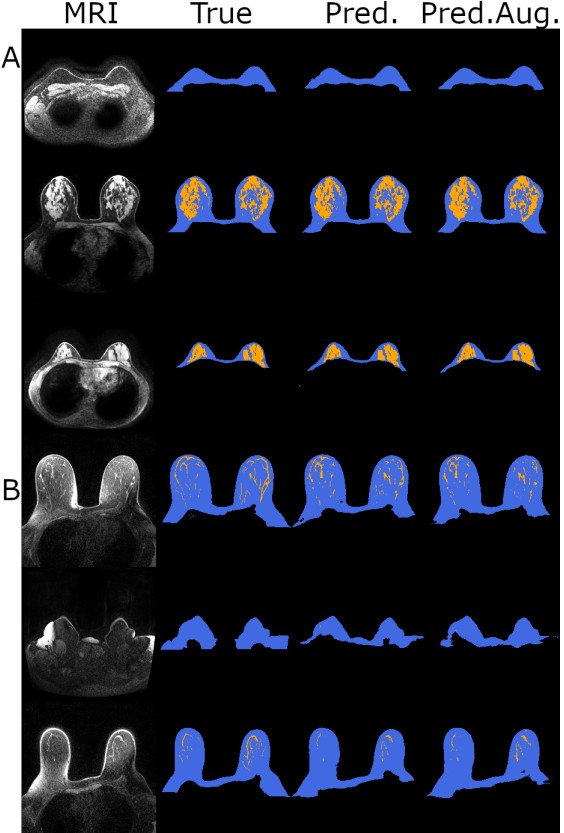

### A.4. Radiological Assessment of Anatomical Changes in Counterfactual Images

The central slice of the 100 volumes used for segmentation were assessed by breast radiologists. The radiologist compared both original and corresponding counterfactual images and classified the counterfactual images into: no, minimal, moderate, significant, and severe anatomical changes. When anatomical changes were observed the radiologist indicated whether such changes were present in the breast tissues or in non-breast tissues. Results of this classication are shown in Table 6.

It is possible to observe that the majority of counterfactual images did not introduce anatomical changes, with just two cases presenting minimal anatomical changes, all of which outside breast tissues.

Table 6: Radiological assessment of anatomical changes in counterfactual images for 100 patients used in the segmentation task and location such changes obtained. NA: not applicable.

| Anatomical Change? | Count | Changes within breast tissues | Changes outside breast tissues |
|---|---|---|---|
| **No** | 98 | NA | NA |
| **Minimal** | 2 | 0 | 2 |
| **Moderate** | 0 | NA | NA |
| **Significant** | 0 | NA | NA |
| **Severe** | 0 | NA | NA |

## A.5. Evaluation Metrics Formulas

$$\text{Top-1 } Accuracy = Accuracy = \frac{TP + TN}{TP + TN + FP + FN} \tag{3}$$

where TP, TN, FP, and FN are the number true positives, true negatives, false positives, and false negatives, respectively.

$$\text{MSE}(y, \hat{y}) = \frac{\sum_{i=0}^{N-1}(y_i - \hat{y}_i)^2}{N} \tag{4}$$

where $y_i$, $\hat{y}_i$ represent the true and predicted values, and $N$ the number of data points.

$$\text{FID} = |\mu_1 - \mu_2| + \text{Tr}(\sigma_1 + \sigma_2 - 2\sqrt{\sigma_1 * \sigma_2}) \tag{5}$$

where $\mu_1$ and $\mu_2$, and $\sigma_1$ and $\sigma_2$ represent the mean and covariance of the two distributions of feature vectors. We used use the RadImageNwt pretrained on medical datasets from MONAI, instead of activations of the pool_3 layer of an Inception v3 pretrained with Imagenet.

$$SSIM(x, y) = \frac{(2\mu_x\mu_y + c_1)(2\sigma_{xy} + c_2)}{(\mu_x^2 + \mu_y^2 + c_1)(\sigma_x^2 + \sigma_y^2 + c_2)} \tag{6}$$

where $\mu_x$ is the pixel sample mean of $x$, $\mu_y$ is the pixel sample mean of $y$, $\sigma_x^2$ is the sample variance of $x$, $\sigma_y^2$ is the sample variance of $y$, $c_1 = (k_1 L)^2$ and $c_2 = (k_2 L)^2$ are two variables to stabilize the division with weak denominator, $L$ is the dynamic range of the pixel-values (typically this is $2^{\#bits \ per \ pixel} - 1$), and $k_1 = 0.01$ and $k_2 = 0.03$ by default.

$$\text{MMD}(\mathbb{F}, X, Y) := \sup_{f \in \mathbb{F}}\left(\frac{1}{m}\sum_{i=1}^{m} f(x_i) - \frac{1}{n}\sum_{j=1}^{n} f(y_j)\right) \tag{7}$$

$$Dice \ Similarity \ Score = \frac{2TP}{2TP + FP + FN} \tag{8}$$

where TP, FP, and FN are the number true positives, false positives, and false negatives, respectively.

$$\text{Counterfactual Prediction Gain} = \mathbb{E}[C(x_i^{\text{cf}}) - C(x_i)] \tag{9}$$

where $C$ is the target classifier and $x_i$ denotes the data point for which a counterfactual $(x_i^{\text{cf}})$ is sought through the proposed cDDGM, which is used to reconstruct $x_i^{\text{cf}}$ (Nemirovsky et al., 2020). The expectations are computed using the test sets.

### A.6. Data, Models' Weights and Code

Derived data, obtained from the Duke-Breast-Cancer-MRI dataset (Saha et al., 2018), and models' weights are made available at https://zenodo.org/records/13495922. Code is available at https://github.com/pedromorao/Counterfactual-MRI-Data-Augmentation.

