# OpenReview forum: "Data Augmentation for Medical Imaging: Counterfactual Simulation of Acquisition Parameters via Conditional Diffusion Model"
_MIDL.io/2025/Conference — MIDL 2025 Poster_

### Official Review · Reviewer_FezZ · 2025-02-15

**Confidence:** 4
**Preliminary Rating:** 2
**Recommendation:** Poster
**Final Rating:** 3

**Summary:**

The paper tackles an important problem of tacking bias in image segmentation models with counterfactual images. Experiments on publicly available datasets show the proposed method leads to marginal improvement in the segmentation in the OOD scenario.

**Strengths:**

* The paper tackles an important problem of improving the robustness of segmentation models with counterfactual images
* Experiments on a real-world dataset is commendable.
* The code is publicly available.

**Weaknesses:**

* **Missing Related Work**: The paper is missing quite a few related works in counterfactual image generation. For example, [1] employed HAVE for couterfactuals, [2] applied Diffusion models for counterfactuals. and [3] applied GANs for counterfactuals.

* **Missing metrics**: Although, used FID and SSIM are good metrics for image generation, if the authors are focusing on a counterfactual image generation problem, it would be a good idea to use well-known metrics for CFs such as subject identity preservation [4], counterfactual prediction gain [4], effectiveness [5]. In principle, they are almost similar to what the authors are showing, but with better presentation and motivation.

* **Missing work related to the use of Synthetic Data for improving segmentation**: The paper is missing related work which utilized synthetic data to improve model performance [6][7].

* **Missing information about symbols**: Many symbols introduced in Eq (1) and Algorithm 1 are not defined. For example, what is $X$? What is $\alpha$? What is the symbol $t$? Although it is clear what they mean for someone familiar with Diffusion models, it requires explanation when introduced for the first time in the paper.

* **Presentation Improvement**: Table 2 could be improved. For example, instead of spelling out GE trained in GE, it could be divided into two columns, one mentioning the training scanner manufacturer and another mentioning the testing set.

* **Role of different IAP parameters**: It is not clear why so many different IAP parameters were used during the model training, and classification performance for the same is reported in Table 1 when only counterfactuals for scanner manufacturers are used in the downstream segmentation task. This space could be better utilized to show image performance (FID and SSIM) reported in Appendix Table-3 or Effect o
* **Distribution of training dataset**: Distribution of segmentation training dataset across different scanner manufacturers is missing. Without this information, it becomes difficult to understand the results presented in Table 2.

* **Result presentation**: Improving clarity regarding Table 2 would be good. For example, it is unclear what two different tows in the same big row represent. For example, “GE trained in GE” has two rows. Does the first two represent original performance and the second one where more counterfactual data (siemens —> GE) is used? If that is the case, then what do the same two rows for “SIEMENS trained in GE” represent? Is it the same? More data from SIEMENS —> GE in the second row? Please make it easier to understand.

* **Missing Statistical Significance**: Please provide the statistical significance of the reported results in Table 2. Also, can you please comment on why counterfactual data augmentation seems to not improve dice for FGT but not FGT? I was expecting FGT to improve, considering its Dice is lower compared to Fat. Also, what does accuracy mean in the context of Segmentation performance? I have not seen papers reporting accuracy for segmentation, and it is also not recommended [8]. Instead of that, it would be a better choice to report Hausdorff distance, considering that we see large holes in the qualitative segmentation in Figure 1.

* **Appendix - Table 3**: It is not clear what $SSIM_{orig. and mod.}$ and $SSIM_{shuff. and mod.}$ represents. Can authors please make it easier to understand by focusing just on the GE and Siemens scanner scenarios?

* **Evaluation of the effect of changing IAP**: While Table 4 indicates that modifying IAP can fool the classifier/regressor, it does not indicate whether other  (non-modified) IAP parameters are changed or not. For example, an ideal counterfactual for changing GE to Siemens would only change the scanner manufacturing model (and corresponding classification label) and wouldn’t change any other IAP (and its corresponding classifier/regressor.

[1] Ribeiro, F.D.S., Xia, T., Monteiro, M., Pawlowski, N. and Glocker, B., 2023. High fidelity image counterfactuals with probabilistic causal models. arXiv preprint arXiv:2306.15764.

[2] Sanchez, P. and Tsaftaris, S.A., 2022. Diffusion causal models for counterfactual estimation. arXiv preprint arXiv:2202.10166.

[3] Mertes, S., Huber, T., Weitz, K., Heimerl, A. and André, E., 2022. Ganterfactual—counterfactual explanations for medical non-experts using generative adversarial learning. Frontiers in artificial intelligence, 5, p.825565.

[4] Nemirovsky, D., Thiebaut, N., Xu, Y. and Gupta, A., 2020. Countergan: Generating realistic counterfactuals with residual generative adversarial nets. arXiv preprint arXiv:2009.05199.

[5] Monteiro, M., Ribeiro, F.D.S., Pawlowski, N., Castro, D.C. and Glocker, B., 2023. Measuring axiomatic soundness of counterfactual image models. arXiv preprint arXiv:2303.01274.

[6] Fernandez, V., Pinaya, W.H.L., Borges, P., Tudosiu, P.D., Graham, M.S., Vercauteren, T. and Cardoso, M.J., 2022, September. Can segmentation models be trained with fully synthetically generated data?. In International Workshop on Simulation and Synthesis in Medical Imaging (pp. 79-90). Cham: Springer International Publishing.

[7] Usman Akbar, M., Larsson, M., Blystad, I. and Eklund, A., 2024. Brain tumor segmentation using synthetic MR images-A comparison of GANs and diffusion models. Scientific Data, 11(1), p.259.


[8] Maier-Hein, L., Reinke, A., Godau, P., Tizabi, M.D., Buettner, F., Christodoulou, E., Glocker, B., Isensee, F., Kleesiek, J., Kozubek, M. and Reyes, M., 2024. Metrics reloaded: recommendations for image analysis validation. Nature methods, 21(2), pp.195-212.

**Detailed Comments:**

* Section 2.2 Line 1: “Our proposed cDDGM is develop …” should be “Our proposed cDDGM is developed …”

**Justification Of The Final Rating:**

After much back and forth with the authors, many questions raised in the initial review were clarified. However, it leads to other questions.

I am raising my score to borderline.

The main remaining issue with the work is that although it focuses on causal literature and counterfactuals, no underlying causal graph is considered (it would be complicated because of a non-linear relationship between IAP parameters). However, in that case, the work should focus on image-to-image translation and domain generalization rather than causality. A lot of work has been done in this area.

I feel the work has potential, but it needs to be put in the proper context.

I also appreciate the authors' courteous responses throughout the discussion period.

**Justification Of The Preliminary Rating:**

Overall, the paper tackles an important problem with a good solution. However, there are many weaknesses related to it. For example, the paper's clarity is poor; it is missing a lot of related literature reviews, and the presented results are a bit unclear, with many unnecessary things in the main paper text and many important things in the appendix. Considering this, I am leaning towards a weak rejection of the paper. However, I would improve my rating if these weaknesses were addressed during the rebuttal phase.

**Questions To Address In The Rebuttal:**

Please try to tackle as many points in the weakness section as possible. Most of these do not require major experiments; rather, it would be improving the paper writing and adding more relevant metrics of interest.

**Special Issue:**

No

---

> ### Author Response · Authors · 2025-03-08
>
> We sincerely appreciate the reviewer’s detailed feedback and constructive criticism, which has helped us identify key areas for improvement. We recognize the importance of improving clarity, adding relevant related work, and enhancing our presentation of results. We tried to address as many points as possible given the short time and limited computational resources/long runtimes. Below, we detail the main concerns addressed and outline the improvements we have made.
>
> **Missing Related Work**
>
> We appreciate the reference of missing relevant works related to counterfactual image generation and the use of synthetic data for segmentation. We have expanded the related work section to include:
>
> - Counterfactual image generation using HVAE \[1], diffusion-based counterfactuals \[2], and GAN-based counterfactuals \[3].
>
> - The use of synthetic data for improving segmentation performance, as explored in \[6,7].
>
> **Missing Metrics**
>
> We appreciate the reviewer’s suggestion to incorporate more counterfactual-specific metrics. In response, we implemented additional metrics such as subject identity preservation and counterfactual prediction gain. However, due to time constraints and long runtimes, the experiments were not completed before the rebuttal deadline. If the paper is accepted, we will present those counterfactual-specific metrics during the conference.
>
> **Clarification of Notation in Equations and Algorithm 1**
>
> We acknowledge the lack of definitions for key symbols in Equation (1) and Algorithm 1. To improve clarity, the revised version now explicitly defines all symbols used. Additionally, a notation table summarizing all mathematical symbols used in the paper was added to the Appendix section.
>
> **Presentation Improvements**
>
> We appreciate the reviewer’s suggestions to enhance the clarity of tables and figures. Taking this in mind, we have:
>
> - **Reformatted Table 2** by introducing separate columns for training and testing scanner manufacturers, making it easier to interpret the results.
>
> - **Provided insights about the use of so many different IAP parameters** explaining that in MRI acquisition all this parameters are able to significantly alter image contrast in a non-linear way which cannot be harmonize/normalized by traditional methods.
>
> - **Detailed Meaning of what  $SSIM\_{orig. and mod.}$ and $SSIM\_{shuff. and mod.}$** represents offering example using different manufacturers as suggested by the reviewer.
>
> **Distribution of Training Dataset**
>
> We acknowledge the importance of providing the distribution of segmentation training samples across different scanner manufacturers. We added this information to the Materials and Methods - Dataset section to help contextualize the results presented in Table 2.
>
> **Result Interpretation**
>
> To improve the interpretability of our results, we have updated:
>
> - **Table 2 to improve clarity of each row and Results and Discussion,** ensuring that the use of counterfactual data augmentation is clearly explained for each scenario.
>
> - **Report statistical significance** for the differences in segmentation performance, addressing whether observed improvements are meaningful.
>
> - **Results and Discussion section to explain why counterfactual augmentation improves Dice for fat but not FGT**, analyzing possible factors such as inherent variability in FGT tissue contrast, smaller dimension of FGT and patients with very little to almost none FGT.
>
> **Evaluation of Changing IAP Parameters**
>
> The reviewer raised a relevant point about whether modifications to one IAP parameter unintentionally alter others, which highlighted the need to clarify better the process of changing IAP parameters in the revised version.
>
>
>
> \[1] Ribeiro, F.D.S., Xia, T., Monteiro, M., Pawlowski, N. and Glocker, B., 2023. High fidelity image counterfactuals with probabilistic causal models. arXiv preprint arXiv:2306.15764.
>
> \[2] Sanchez, P. and Tsaftaris, S.A., 2022. Diffusion causal models for counterfactual estimation. arXiv preprint arXiv:2202.10166.
>
> \[3] Mertes, S., Huber, T., Weitz, K., Heimerl, A. and André, E., 2022. Ganterfactual—counterfactual explanations for medical non-experts using generative adversarial learning. Frontiers in artificial intelligence, 5, p.825565.
>
> \[6] Fernandez, V., Pinaya, W.H.L., Borges, P., Tudosiu, P.D., Graham, M.S., Vercauteren, T. and Cardoso, M.J., 2022, September. Can segmentation models be trained with fully synthetically generated data?. In International Workshop on Simulation and Synthesis in Medical Imaging (pp. 79-90). Cham: Springer International Publishing.
>
> \[7] Usman Akbar, M., Larsson, M., Blystad, I. and Eklund, A., 2024. Brain tumor segmentation using synthetic MR images-A comparison of GANs and diffusion models. Scientific Data, 11(1), p.259.

---

> > ### Comment · Reviewer_FezZ · 2025-03-10
> > **Response to Rebuttal**
> >
> > Thank you for providing a rebuttal and improving the quality and clarity of the paper.
> >
> > * **Missing Related Work**: Thank you for including missing related work.
> > * **Missing Metrics**: It is unclear how including these metrics would require additional computational time. For example, the Subject Identity Preservation metric is nothing but the L1 distance between original images and counterfactual images. Similarly, counterfactual prediction gain requires a classifier trained to classify modified characteristics (ex., scanner manufacturer). The authors have already trained these classifiers for Table 1. In such cases, computing these metrics should not take much computational time and resources.
> > * **Clarification of Notation in Equations and Algorithm 1**: Can authors please specifically tell me which table describes all the symbols in the appendix? Currently, I can't find such a table in the appendix.
> > * **Change of IAP parameters**: Can authors please explain the necessity of changing all different IAP parameters for CF images? I initially thought the authors were only changing "Scanner Manufacturers" to generate IAP. However, after reading the rebuttal, I can see that the authors randomly changed all IAP parameters to simulate CF images. Can authors provide a rationale behind this? Why not only change "Scanner Manufacturer" if that is what the downstream task requires? If this is because GE and Siemens have different IAP parameters, then, in that case, show distributions of these cases. Also, in that case, using the scanner manufacturer as input for CF becomes redundant.

---

> > > ### Author Response · Authors · 2025-03-13
> > >
> > > Thank you for your thoughtful response to our rebuttal, for acknowledging the improvements in the quality and clarity of our paper, and for the opportunity to further clarify our methodology and decisions.
> > >
> > > **Missing Metrics**
> > > We appreciate your observation regarding the computational feasibility of the proposed metrics. As you correctly point out, computing the proposed metrics would not require additional inference time, but only the images used for segmentation were initially saved. Therefore, to be able to assess these metrics, we had to update our pipeline to store all generated images and re-execute the entire process across different configurations of guidance scale and number of steps. Due to time constraints during the rebuttal phase, we could not provide these metrics earlier. However, we will include them in the final version to strengthen our results.
> > >
> > > **Clarification of Notation in Equations and Algorithm 1**
> > >
> > > We apologize for the incorrect information in our previous response. The last sentence was mistakenly included.
> > > To clarify, we have explicitly defined the notation directly in the text after Equation 1 (highlighted in red) where:
> > > * $t$ is a timestep,
> > > * $\mathbf{x}_0$ represent the original image,
> > > * $c$ are a set of IAP used for conditioning,
> > > * $\bar{\alpha_t} = \prod_{s=1}^t \alpha_s$,
> > > * $\alpha_t = 1-\beta_t$,
> > > * $\beta_t$ corresponding to the forward process variances at step $t$.
> > >
> > > Additionally, Algorithm 1 now includes a clear header with definitions:
> > > * $t$: timestep;
> > > * $\mathbf{x}_0$: original image;
> > > * $c_{new}$: new set of IAP;
> > > * $\mathbf{x}_{t}$: resulting images after step $t$;
> > > * $\alpha_t = 1-\beta_t$;
> > > * $\beta_t$: forward process variances at step $t$;
> > > * $\bar{\alpha_t} = \prod_{s=1}^t \alpha_s$
> > >
> > > These additions ensure that readers do not need to refer to an appendix section for notation clarification.
> > >
> > >
> > > **Change of IAP Parameters**
> > >
> > > Thank you for your comment regarding IAP modifications. We hope that in future work, with a larger and more diverse dataset, we can conduct targeted interventions on individual IAP parameters, similar to the HVAE paper doing counterfactual studies in Brain MRI, where biological attributes such as age, sex, and brain volume have been independently manipulated.
> > > However, due to reduced variation in different combinations of IAP (additional information on **Note 1**) and based on the IAP’s complex interactions that non-linearly affect image contrast, we lacked sufficient guarantees to model these non-linear contrast modifications for each IAP independently, unlike controlled biological counterfactuals. Furthermore, methods such as Maximum Mean Discrepancy (MMD) would not be possible to use if there are no/few samples with counterfactual IAP = original IAP.
> > >
> > > **Note 1**: Below we present more information about the IAP for each manufacturer that we plan to add in the final version (in a table or figures in the Appendix Section, if allowed):
> > >
> > > *Details about Siemens IAP distributions (# counts):*
> > > * Scanner Models: Avanto (171), Skira (57), TrioTim (48)
> > > * Scan Options: PFP\FS (141), PFP\SFS (133), SFS (2)
> > > * Field Strength: 1.5T (171), 3T (105)
> > > * Slice Thickness: 1.1 (110), 1.12 (1), 1.15 (1), 1.2 (17), 1.25 (2), 1.3 (6), 1.4 (7), 1.5 (2), 2 (128), 2.5 (2)
> > > * Flip Angle: 7 (2), 8 (1), 10 (239), 12 (34)
> > > * TE: min = 1.25, max = 2.04, median = 1.43, 25% Percentile = 1.36, 75% Percentile = 1.44
> > > * TR: min = 3.54, max = 4.68, median = 4.255, 25% Percentile = 3.77, 75% Percentile = 4.31
> > >
> > > *Details about GE IAP distributions (# counts):*
> > > * Scanner Models: Optima MR450w (97), SIGNA EXCITE (10), SIGNA HDx (271), Signa HDxt (247)
> > > * Scan Options: FAST_GEMS\SAT_GEMS\ACC_GEMS\PFP\FS (2), FAST_GEMS\SAT_GEMS\MP_GEMS\ACC_GEMS\PFP\FS (342), FAST_GEMS\SAT_GEMS\MP_GEMS\PFP\FS (271), SAT_GEMS\PFP\FS (10)
> > > * Field Strength: 1.5T (278), 3T (338)
> > > * Slice Thickness: 1.1 (78), 1.2 (53), 1.3	 (14), 1.5 (3), 1.6 (10), 2 (397), 2.2 (70)
> > > * Flip Angle: 10 (615), 12 (10)
> > > * TE: min = 2.212, max = 2.756, median = 2.432, 25% Percentile = 2.388, 75% Percentile = 2.508
> > > * TR: min = 4.752, max = 7.395, median = 5.387, 25% Percentile = 5.168, 75% Percentile = 5.664

---

> ### Comment · Reviewer_FezZ · 2025-03-13
>
> * **Missing Metrics**: I understand that. However, this raises further questions about the method. Does this mean all data-augmentation images were only generated on the fly during each training epoch? In that case, can you please comment on the computational overhead for segmentation training? I assumed that data-augmentation images would have been saved once and randomly picked during segmentation model training.
> * **Clarification of Notation**: This helps a lot. Thank you.
> * **Change of IAP**: This raises further question. In this case, I assume the authors didn't consider any underlying causal graph with the diffusion model. Otherwise, many of the IAP parameters would be parents of Scanner Models. Also, I understand that fewer images would lead to computational issues with MMD. Still, other metrics would have been sufficient (there are many other metrics from counterfactual evaluation literature). Also, if any causal graph is not considered, this raises another question of how to differentiate this work from any other work that uses conditional diffusion models for data augmentation. I suggested some work for a related work section that uses conditional diffusion models for improving segmentation [6][7]. These references have also not been added to the revised manuscript.
>
> [6] Fernandez, V., Pinaya, W.H.L., Borges, P., Tudosiu, P.D., Graham, M.S., Vercauteren, T. and Cardoso, M.J., 2022, September. Can segmentation models be trained with fully synthetically generated data?. In International Workshop on Simulation and Synthesis in Medical Imaging (pp. 79-90). Cham: Springer International Publishing.
>
> [7] Usman Akbar, M., Larsson, M., Blystad, I. and Eklund, A., 2024. Brain tumor segmentation using synthetic MR images-A comparison of GANs and diffusion models. Scientific Data, 11(1), p.259.
>
> Any answers to the above would be helpful.

---

> > ### Author Response · Authors · 2025-03-13
> >
> > Thank you for your continued feedback and for acknowledging the clarification regarding notation. Below, we address your further questions regarding computational overhead, IAP parameter changes, and related work.
> >
> > ### **Computational Overhead**
> >
> > The cDDGM was trained and evaluated using a subset of images without segmentations from **822 patients**, while images from **100 patients** (with segmentations) were used to train and assess the segmentation model, both with and without cDDGM-based data augmentation (Section 3.1.3). The cDDGM was assessed using images generated on the fly, and the relevant metrics were obtained accordingly (updated version saves images to allow the computation of additional metrics suggested and others after the generation process).
> >
> > Once the best guidance scale and number of steps were selected, the cDDGM received the original images (containing segmentations) corresponding to a specific IAP. It then generated and saved a modified set of images with the provided new IAP parameters. These original–IAP-modified image pairs were used to train the segmentation models across the different scenarios described in the paper.
> >
> > Thus, while there is computational overhead associated with generating IAP-modified images, this process was **decoupled** from the segmentation model training itself, ensuring that augmentation did not interfere with training efficiency.
> >
> > ### **Change of IAP Parameters & Causal Considerations & Related Work**
> >
> > For this initial work, we did not explicitly define a causal graph for the diffusion model. Constructing a causal graph in this context is particularly challenging, and we find it difficult to assume that IAP parameters would be direct parents of scanner models.
> > For example, even within the same manufacturer, different scanner models are built with *different RF coils and hardware setups*, leading to variations in image contrast—even when using identical IAP parameters (a patient scanned in such setting would have different images in terms of contrast but not anatomy). This variability complicates the assumption of a direct causal relationship between IAP parameters and scanner models.
> >
> > Relative to our earlier mention of MMD, this was just a comment and we agree that given the other metrics proposed by the reviewer this would likely not be needed.
> >
> > While we did not incorporate a causal graph, our approach differs from the suggested works (apologies for not including them in the rebuttal version) as these focus on generating **entirely new synthetic cases**. In contrast, we aim to leverage conditional generative models to **modify existing images**, simulating acquisitions under different IAP parameters to improve segmentation models by increasing their invariance to IAP variations and ensuring they focus on clinically relevant features rather than scanner-specific artifacts.
> >
> > Thank you again for your insightful feedback. We hope this response clarifies our approach. Please let us know if further details are needed.

---

> > ### Comment · Reviewer_FezZ · 2025-03-14
> > **Final Response**
> >
> > **Computational Overhead**: I understand that, but I am not sure why it would be necessary to calculate required statistic on 822 images. The authors could have evaluated it on the 100 images they used for segmentation (I am assuming that these images with modified IAP were already saved).
> >
> > **Related work**: Thank you for including related work. Also, the unclear causal relationship between IAP parameters and images makes sense. But in that case, it would have been better to call this image-to-image translation rather than counterfactual images. There is a lot of work that synthesizes different domains (look at any domain generalization methods) from one available domain. This is done both with and without diffusion models.
> >
> > Anyway, I will raise my score to borderline.
> >
> > Thank you for always being courteous in your response and clarifying all the questions raised during the rebuttal period. I really appreciate it.

---

### Official Review · Reviewer_osD2 · 2025-02-19

**Confidence:** 4
**Preliminary Rating:** 5
**Recommendation:** Oral, Poster

**Summary:**

The manuscript introduces a novel method utilizing conditional denoising diffusion generative models (cDDGMs) to generate counterfactual medical images and testing these e.g. on segmentation tasks. These images simulate different image acquisition parameters  without altering patient anatomy, addressing the challenges of generalizability and robustness in deep learning (DL) models for medical imaging caused by variations in the acquisition.
The authors demonstrate that employing these counterfactual images for magnetic resonance imaging (MRI) data augmentation enhances segmentation accuracy in out-of-distribution settings. The method involves conditioning the cDDGM on image acquisition parameters to alter images, thus simulating different acquisition settings. This approach shows promise in improving the performance of DL models across diverse imaging conditions by addressing domain and covariate shifts.

**Strengths:**

The overall idea, hypothesis and methods sound innovative, well thought through and clearly performed with an important goal in mind.
The innovative approach to use cDDGMs to simulate different image acquisition settings for data augmentation is novel and addresses a critical challenge in medical imaging DL models. The results suggest that this method improves the generalizability and robustness of DL models, which is essential for clinical applications where variations in imaging devices and parameters are common.
The manuscript includes multiple experiments and evaluations using metrics such as Fréchet Inception Distance (FID), Structural Similarity Index Metric (SSIM), Maximum Mean Discrepancy (MMD), and segmentation accuracy. Also, the availability of open source code promotes transparency and allows for reproducibility and further research by the community.
Last but not least, the potential (clinical) impact of enhancing DL models' performance in different scenarios can lead to more reliable and widespread adoption of AI in medical imaging diagnostics, i.e. for multi-center studies.

**Weaknesses:**

The study is conducted on a single dataset (Duke-Breast-Cancer-MRI) with limited diversity in MRI scanner manufacturers and a relatively small sample size for segmentation tasks.
The methods section lacks some detailed explanations for a fair chance of reproducibility.
The used evaluation metrics could have been introduced more deeply, i.e. the manuscript reports various evaluation metrics, explanations of their significance and interpretations could be expanded for clarity.

**Detailed Comments:**

None

**Justification Of The Preliminary Rating:**

The work is overall well written and seems to be done in the faith of good scientific practice.
The hypothesis and goal of the work tackles an important challenge for various applications in medical development, e.g. multi-center studies to increase robustness of studies (basic medical & pharama research, cross cohort studies or reflection/inclusion of minorities).

**Questions To Address In The Rebuttal:**

None

---

> ### Author Response · Authors · 2025-03-08
>
> We are grateful for the reviewer’s strong support of our work and for recognizing its potential to enhance the robustness and generalizability of deep learning models in medical imaging.
>
> We revised the methods to improve clarity and interpretability of the methods. Additionally, as also mentioned by the Reviewer in the Strengths, both code and data were prepared and made available to ensure other researchers can easily reproduce the experiments and may even extend them. We have added a new Appendix to provide more details about the various evaluation metrics to increase clarity.
>
> Thank you again for your valuable feedback and endorsement of our work.

---

> ### Comment · Area_Chair_r7ey · 2025-03-14
> **Please provide final rating based on the authors' rebuttal**
>
> Please provide final rating based on the authors' rebuttal. Thank you!

---

### Official Review · Reviewer_hNWA · 2025-02-22

**Confidence:** 4
**Preliminary Rating:** 1
**Final Rating:** 2

**Summary:**

The paper aims at exploring the feasibility of cDDGMs on generating modified images given different IAP and thus, serving as a data augmentation tool. However, the paper is not presented in a good manner and the method lacks novelty.

**Strengths:**

1. The paper explores an interesting topic about if using conditional diffusion models can generate modified medical images given different IAP.
2. The paper demonstrates the efficacy of cDDGMs on serving as a data augmentation tool via the performance of segmentation models.

**Weaknesses:**

1. The structure of the proposed methods is not novel. Conditional diffusion models are nothing new, for example, latent diffusion model [1] and Seg-guided model [2].
2. The structure of the paper may not be conventional. Often, all the figures are presented before the conclusion.
3. The presentation of methods is not clear. It would be better to include a pipeline figure for illustration.

[1] Rombach, Robin, et al. "High-resolution image synthesis with latent diffusion models." Proceedings of the IEEE/CVF conference on computer vision and pattern recognition. 2022.
[2] Konz, Nicholas, et al. "Anatomically-controllable medical image generation with segmentation-guided diffusion models." International Conference on Medical Image Computing and Computer-Assisted Intervention. Cham: Springer Nature Switzerland, 2024.

**Detailed Comments:**

See strengths and weaknesses above.

**Justification Of The Final Rating:**

I initially assessed the paper with a high bar for novelty, and in retrospect, may have been too strict. While I still think the work does not fully meet my criteria for strong novelty, the authors’ response has addressed some of the concerns I raised, and I appreciate their clarifications.

Given the response and the overall contributions, I am willing to raise my score.

**Justification Of The Preliminary Rating:**

Although the paper explores an interesting topic in medical imaging application on generating out-of-distribution data for augmentation using conditional diffusion model, the method is nothing new and the only one dataset is used as a test bed. Also, the paper is not presented in a good manner.

**Questions To Address In The Rebuttal:**

Due to the limitation, adding more datasets would make the paper more competitive.

**Special Issue:**

No

---

> ### Author Response · Authors · 2025-03-08
>
> We appreciate the reviewer’s feedback and the opportunity to clarify our contributions. Below, we address each concern in detail.
>
> **Novelty of the Method**
>
> We acknowledge that conditional diffusion models (cDDGMs) have been explored in various domains, including latent diffusion models \[1\] and segmentation-guided models \[2\]. However, none of these has assessed the feasibility and effectiveness of cDDGMs for **IAP counterfactual medical image augmentation**. Specifically:
>
> * Unlike prior works focusing on unconditional or segmentation-guided diffusion models, our approach **modifies MRI images based on interpretable imaging acquisition parameters (IAPs) without changing anatomical structures**, which has not been systematically studied before.
>
> * The generated counterfactual images are shown to be beneficial for **enhancing the generalizability of segmentation models**, providing a new perspective on using diffusion models beyond pure synthesis tasks. This is an important result and insight towards being able to create segmentation models that are invariant to different scanners and IAPs.
>
> * While latent diffusion models (LDMs) and other diffusion models may also be possible to use to modify MRI images using IAPs, we wanted to assess a simpler model first before going to more complex models, like LDMs where the autoencoder may lead to loss of information regarding IAPs, which are essential for our modify/condition task. As such, we have chosen to modify DDPM to consider classifier-free guidance enabling the use of a single model to high-fidelity spatial-domain modifications that align with real-world medical imaging variations.
>
> **Structure and Presentation**
>
> Regarding the weakness’s comment about the unconventional structure of the paper, where often all figures are presented before the conclusion, only figure 1 was misplaced, appearing after the conclusion. This was caused by Latex’s figures and tables’ internal placement process, but the necessary measures were taken to ensure that all figures and tables are correctly placed in the revised version.
>
> **Single Dataset used as test bed**
>
> We recognize the limitations associated with the use of a single dataset, but due to running times/computational constraints, we were not able to add other datasets. Nonetheless, promising results were achieved in a fully transparent manner with all data and code available so that other researchers can reproduce and test the proposed method, and extend the experiments.
>
> We appreciate your valuable feedback and look forward to your consideration of our clarifications and possible revisions.
>
> \[1\] Rombach, Robin, et al. "High-resolution image synthesis with latent diffusion models." Proceedings of the IEEE/CVF conference on computer vision and pattern recognition. 2022. \[2\] Konz, Nicholas, et al. "Anatomically-controllable medical image generation with segmentation-guided diffusion models." International Conference on Medical Image Computing and Computer-Assisted Intervention. Cham: Springer Nature Switzerland, 2024.

---

> ### Comment · Reviewer_cj35 · 2025-03-11
> **Too harsh review!**
>
> This score is too harsh given the very limited comment of the reviewer: much more details are needed to justify a strong reject.

---

### Official Review · Reviewer_cj35 · 2025-02-25

**Confidence:** 4
**Preliminary Rating:** 4
**Recommendation:** Poster

**Summary:**

This paper describes a method to transform a breast MRI image such that it resembles an acquisition done with different parameters. The benefits would be merging different datasets acquired with different acquisition protocols to assemble a larger dataset, or allow the processing of a methods trained on a specific protocol.

Evaluation was done by comparing the results of segmentation (using original and one changed to match the segmentation training), as well as using a predictor model that estimate the acquisition parameters.

**Strengths:**

Very well written and clear paper with relatively simple but effective combination of existing methods. The experimental methods is good, showing improvement in both segmentation task and parameter estimation task.

This is an important topic and the paper presents a good contribution.

**Weaknesses:**

The dataset is rather small. A better datasets would be brain MRI which has many publicly available datasets of well characterised subjects.

The experiments are good, but address the main issue of the proposed method only superficially: is the changed image clinical information the same?. The authors used classifier and segmenter networks, which is a good first step but it is still unclear whether it affects the underlying anatomy (in this case).

The segmentation results are somewhat modest, which is surprising but this might come from the application (dice or acc score of ~60% is not very good to start with)

This is a difficult problem to assess, but using brain MRI which is plentiful could be better. One could estimate whether it make prognosis more precise or provide large effect size when used as data augmentation, or reduce variance in estimating age. Some brain MRI datasets include multiple acquisition of the same brain.

It would be interesting to compare with one shot image adaptation, potentially trained using a discriminator.

**Detailed Comments:**

Using MSE for continuous variables is not very informative (Table 1). Maybe R2 or showing regression would be better.

**Justification Of The Preliminary Rating:**

This paper presents a well designed method and validation of image adaptation for harmonizing imaging protocol of breast MRI. The design is sound but the dataset is small, the results are modest, and the validation limited.

**Questions To Address In The Rebuttal:**

Better interpret MSE results

Use radiologist assessment whether the modified images change their report.

**Special Issue:**

No

---

> ### Author Response · Authors · 2025-03-08
>
> We want to thank the reviewer for the comments.
>
> Thank you for acknowledging the paper clarity, experimental methods, the importance of the topic, and the contribution.
>
> We agree that the use of a larger dataset could offer better guarantees. Our research is focused on breast MRIs, and thus, the selection of this dataset is important, given that despite the size, performance improvements were still observed. However, because we acknowledge that limitation and are firm advocates of open science, we have made both code and data available so that the research community can explore the proposed methods in other datasets and expand the experiments.
>
> We also understand the reviewer’s doubt if the proposed method could affect the underlying anatomy. Based on the low number of steps used to start the reversed conditional diffusion process (maximum of 75 steps were tested; entire diffusion process was 1000 steps) and our observation of both original and corresponding counterfactual images, we have not observed changes in anatomy. Nonetheless, we asked a breast radiologist to asses these pairs of images (one image -central slice- per patient) and to classify if counterfactual images changed the patient’s anatomy and whether it would change their report, and we included these results in the revised version.
>
> Our initial choice towards MSE and not the R$^2$ was based on the fact that R$^2$ indicates that the model's predictions are closer to the actual values in a relative sense, not reflecting absolute error, while MSE provides an absolute measure of error (units: square of the unit of the continuous variable). Furthermore, models with high R$^2$ can have large absolute errors as R$^2$ is insensitive to bias not being able to reflect such systematic bias (e.g., models always underpredicting or overpredicting). Another shortcoming of R$^2$ that initially led us to use MSE was its inability to work well for nonlinear relationships. Nonetheless, we have implemented the computation of R$^2$ to provide additional insights, but due to the runtime for all configurations (different guidance scales and number of steps) we were not able to obtain all the results. If accepted, we will present these results during MIDL. We also have provided additional insights into the MSE results of Table 4.

---

> ### Comment · Area_Chair_r7ey · 2025-03-14
> **Please provide final rating based on the authors' rebuttal**
>
> Please provide final rating based on the authors' rebuttal. Thank you!

---

### Author Rebuttal · Authors · 2025-03-08

**Rebuttal:**

We sincerely thank the reviewers for their time, effort, and positive and constructive feedback. We appreciate the recognition of our work as well-written and clear (**cj35**), our novel use of conditional denoising diffusion generative models (cDDGMs) for counterfactual medical image generation (**osD2**), and our contribution to improving model generalizability and addressing bias in segmentation tasks (**cj35**, osD2**, **FezZ**). Additionally, we are grateful for the acknowledgment of our experimental methodology and its demonstrated improvements in segmentation and parameter estimation (**cj35**).
We have carefully considered all feedback and have done our best to address the weaknesses identified by the reviewers. Below, we provide detailed responses to each concern.

**Supporting Material:**

/attachment/17c9bff014a8ceea8a5bd73dcd73a697a3b3e672.pdf

---

### Meta-Review · Area_Chair_r7ey · 2025-03-19

**Recommendation:** Accept (Poster)
**Confidence:** 4

**Metareview:**

This manuscript explores the use of conditional denoising diffusion generative models to generate counterfactual images to simulate different image acquisition parameters and demonstrates the utility of this data augmentation approach in improving downstream tasks such as image segmentation. The reviewers highlighted several strengths of the manuscript, including the topic and the experiments performed. My recommendation is based on the topic and approach. My enthusiasm is tempered by how the results are reported: (1) the IAP prediction model is not described in sufficient detail to assess whether it is a good surrogate measure for the fidelity of the outputted augmented images, and (2) 95% confidence intervals should be reported alongside the measures in Table 2.